# Two High-Quality *Cygnus* Genome Assemblies Reveal Genomic Variations Associated with Plumage Color

**DOI:** 10.3390/ijms242316953

**Published:** 2023-11-29

**Authors:** Yuqing Chong, Xiaolong Tu, Ying Lu, Zhendong Gao, Xiaoming He, Jieyun Hong, Jiao Wu, Dongdong Wu, Dongmei Xi, Weidong Deng

**Affiliations:** 1Faculty of Animal Science and Technology, Yunnan Agricultural University, Kunming 650201, China; 2022004@ynau.edu.cn (Y.C.); yinglu_1998@163.com (Y.L.); zander_gao@163.com (Z.G.); xiaominghe@foxmail.com (X.H.); hongjieyun@163.com (J.H.); 15229238680@163.com (J.W.); xidmynau@163.com (D.X.); 2State Key Laboratory of Genetic Resources and Evolution, Kunming Institute of Zoology, Chinese Academy of Sciences, Kunming 650223, China; wudongdong@mail.kiz.ac.cn

**Keywords:** mute swan (*Cygnus olor*), black swan (*Cygnus atratus*), high-quality genome, plumage color, melanogenic pathway

## Abstract

As an exemplary model for examining molecular mechanisms responsible for extreme phenotypic variations, plumage color has garnered significant interest. The *Cygnus* genus features two species, *Cygnus olor* and *Cygnus atratus*, that exhibit striking disparities in plumage color. However, the molecular foundation for this differentiation has remained elusive. Herein, we present two high-quality genomes for *C. olor* and *C. atratus*, procured using the Illumina and Nanopore technologies. The assembled genome of *C. olor* was 1.12 Gb in size with a contig N50 of 26.82 Mb, while its counterpart was 1.13 Gb in size with a contig N50 of 21.91 Mb. A comparative analysis unveiled three genes (*TYR*, *SLC45A2*, and *SLC7A11*) with structural variants in the melanogenic pathway. Notably, we also identified a novel gene, PWWP domain containing 2A (*PWWP2A*), that is related to plumage color, for the first time. Using targeted gene modification analysis, we demonstrated the potential genetic effect of the *PWWP2A* variant on pigment gene expression and melanin production. Finally, our findings offer insight into the intricate pattern of pigmentation and the role of polygenes in birds. Furthermore, these two high-quality genome references provide a comprehensive resource and perspective for comparative functional and genetic studies of evolution within the *Cygnus* genus.

## 1. Introduction

Feather pigmentation variations emerge as a prevalent focal point in the domain of avian biology. The myriad hues of feathers showcased by birds assume critical roles in domains encompassing covert strategies, courtship rituals, nutritional dynamics, and signal recognition [1]. Deciphering the intricate dimensions of pigment deposition in avian plumage holds the promise of yielding profound insights into the complex behavioral patterns, ecological dynamics, and evolutionary adaptations. Moreover, the distinct coloration observed between males and females in polygamous avian species, as well as the parallelism in coloration within their monogamous counterparts, adds an intriguing layer to the study of avian biology. In polygamous birds, the divergence in coloration between genders often reflects adaptations linked to mate selection and reproductive strategies, while females exhibit more subdued tones for enhanced camouflage during nesting. Conversely, monogamous bird species exhibit a different dynamic, where the similarity in coloration between males and females may signify shared parental responsibilities and a cooperative breeding strategy.

Owing to the diverse hues and ease of observation, plumage color provides an outstanding model for exploring molecular mechanisms underpinning exceptional phenotypic variation. Studies on birds have shown that plumage color is regulated by a variety of mechanisms [2,3,4]. Melanin, produced by neural crest cell-derived melanocytes, may be a key factor affecting pigmentation in bird feathers and mammalian hair [5,6]. Previous studies have disclosed the DNA polymorphisms of dozens of genes responsible for melanin-based coloration variations [7,8,9,10]. For instance, the melanocortin-1 receptor (*MC1R*) gene critically governs melanocyte-synthesized melanin type [11,12,13]. Tyrosinase, encoded by the tyrosinase gene, serves as the rate-limiting enzyme affecting melanin production and is exclusively produced by melanocyte cells within the melanosome membrane [8,14]. This enzyme’s functions include oxidizing tyrosine to dihydroxyphenylalanine (DOPA) and determining the synthesized melanin type (eumelanin or phaeomelanin) [15,16]. Additionally, some studies have preliminarily scrutinized and analyzed the genetic mechanisms behind skin or hair color differences in other species. For example, an amino acid substitution (His615Arg) in OCA2 diminishes melanin production, playing a crucial role in East Asians’ convergent skin lightening during recent human evolution [17], with three strongly associated loci (EDNRB, CNTLN, and PINK1) accounting for approximately 20% of the total coat color variance in Diannan small-ear pigs [18], and the genetic basis of red plumage has been reported, involving two genomic regions introgressed from red siskins required for red coloration [19]. These investigations contribute to a deeper understanding of the genetic mechanisms driving phenotypic variations in plumage color and other species’ pigmentation traits, paving the way for further advancements in the field.

The *Cygnus* genus, a member of the *Anserinae*, encompasses *Cygnus olor* (mute swan), *Cygnus atratus* (black swan), *Cygnus melanocorypha* (black-necked swan), *Cygnus buccinator* (trumpeter swan), *Cygnus cygnus* (whooper swan), and *Cygnus columbianus* (tundra swan) [20]. Throughout *Cygnus* evolution, numerous traits have considerably diversified among species, including body morphology and reproductive characteristics. Among these traits, feather coloration is the most conspicuous. Four species inhabiting the Northern hemisphere, namely *C. olor*, *C. cygnus*, *C. buccinator*, and *C. columbianus*, exhibit immaculate white plumage. Conversely, in the Southern hemisphere, *C. atratus* presents entirely black plumage, while *C. melanocorypha* displays white plumage on the torso and wings, accompanied by black feathers on the head and neck. Taking the mute swan (*C. olor*) and the black swan (*C. atratus*) as exemplars, a myriad of distinctions in habitat and species characteristics unfolds. Originating in Europe and Asia, the mute swan is renowned for its resplendent white plumage and distinctive orange carunculated bill. Males typically exhibit an adult weight ranging from 10 to 12 Kg, while their female counterparts weigh from 8 to 10 Kg. Conversely, the black swan, an exclusive denizen of Australia, distinguishes itself through the obsidian sheen of its plumage and the vibrant red hue of its bill. Remarkably lighter in stature, the average adult weight of male black swan hovers around a mere 6 Kg, with females weighing approximately 5 Kg. In positing that the chromatic transition of plumage within the swan genus is governed by the intricate interplay of multiple genes, we grapple with a potentially intricate phenomenon. The complexity arises from the intricate variations and interactions among these genes. Research has shown that the melanic specie, *C. atratus*, possess amino acid alterations at crucial functional sites within the MC1R protein, correlating with heightened MC1R activity and melanism [21].

Nonetheless, the molecular underpinnings of plumage color variation among *Cygnus* species remain elusive due to the absence of high-quality genomic resources. In the current investigation, we de novo assembled the genomes of *C. olor* (exhibiting pristine white plumage) and *C. atratus* (displaying entirely black plumage) using the Illumina and Nanopore methodologies. Via the lens of comparative genomics, our endeavor is directed toward the discernment of genes and variations intricately associated with feather pigmentation. Simultaneously, we embark on an exploration of the intricate molecular underpinnings orchestrating the transitions in plumage hues within swan species.

## 2. Results

### 2.1. Genome Sequencing, Assembly and Annotation

Leveraging the technical benefits of third-generation sequencing’s long read lengths and second-generation sequencing’s single-base accuracy, we adopted a hybrid sequencing strategy using Nanopore and Illumina technologies to establish the reference genomes of *C. olor* and *C. atratus*. A staggering amount of 54.69 and 60.39 Gb of clean Nanopore data, complemented by 65.56 and 62.75 Gb of high-quality Illumina Hiseq data, were harnessed for *C. olor* and *C. atratus*, respectively (Appendix A). Ultimately, the final genome assemblies for *C. olor* and *C. atratus* spanned 1.12 Gb and 1.13 Gb, respectively, boasting contig N50 lengths of 26.82 Mb (*C. olor*) and 21.91 Mb (*C. atratus*). An assessment of the assembly completeness using BUSCO v2.0 demonstrated 98.45% and 98.30% complete single-copy genes in *C. olor* and *C. atratus*, respectively, attesting to the high quality of the genome assemblies (Table 1 and Appendix A).

A total of 86.69 Mb (7.71%) and 93.49 Mb (8.28%) of the genome assembly of *C. olor* and *C. atratus*, respectively, was occupied by the predicted repeat sequences (Appendix A). We adopted a comprehensive approach that integrated de novo prediction and protein homology searches, resulting in 17,759 and 17,835 predicted genes for *C. olor* and *C. atratus*, respectively (Appendix A). Noncoding genes, including 207 miRNAs, 128 rRNAs, and 376 tRNAs for *C. olor*, and 215 miRNAs, 130 rRNAs, and 369 tRNAs for *C. atratus*, were also predicted (Appendix A). Functional annotation of all genes was performed using GO, KEGG, NR, KOG, and TrEMBL databases, and a high proportion of gene sets (95.66% for *C. olor* and 95.59% for *C. atratus*) were functionally annotated (Appendix A).

### 2.2. Phylogenetic Analysis

The elucidation of evolutionary relationships among species and the functionality of genes is of utmost importance, and phylogenetic analysis provides a valuable tool for this purpose. To discern the phylogenetic positions of *C. olor* and *C. atratus*, we retrieved protein sequences of Southern Screamer (*Chauna torquata*), Magpie Goose (*Anseranas semipalmata*), Ruddy Duck (*Oxyura jamaicensis*), Muscovy Duck (*Cairina moschata*), Mallard duck (*Anas platyrhynchos*), Tufted Duck (*Aythya fuligula*), Swan Goose (*Anser cygnoides*), and Pink-footed Goose (*Anser brachyrhynchus*) from public databases, and utilized Jungle Fowl (*Gallus gallus*) as the root for the tree. For *C. olor*, out of 17,759 protein-coding genes, 16,649 genes were grouped into 14,052 orthologue groups, with an average of 1.18 genes per orthologue group. Similarly, for *C. atratus*, 16,694 genes were grouped into 14,037 orthologue groups out of 17,835 protein-coding genes, with an average of 1.19 genes per orthologue group (Figure 1 and Appendix A).

A phylogenetic analysis was conducted and a tree was generated, which is presented in Figure 1 and Appendix A. The results indicate that the two species of *Cygnus*, *C. olor* and *C. atratus*, are closely related to two species of *Anser* (*A. cygnoides* and *A. brachyrhynchus*), and the estimated median divergence time between *Cygnus* and *Anser* was 25.2 million years ago (Mya) (with a range of 20.8–29.3 Mya). Furthermore, the estimated median divergence time between *C. olor* and *C. atratus* was approximately 10.8 Mya (with a range of 7.7–13.8 Mya) (refer to Appendix A for details).

### 2.3. Unique and Expanded Orthologue Groups

The presence of unique and expanded orthologue groups in a genome can indicate lineage-specific functions or pathways. In the case of *C. olor* and *C. atratus*, 3 and 22 unique gene families were identified, respectively. Among these, two known genes were found in *C. atratus*, including the epidermal differentiation protein containing the cysteine histidine motifs 2 (*EDCH2*) gene, which is involved in epidermal differentiation, and the small nuclear ribonucleoprotein polypeptide E (*SNRPE*) gene, which is involved in RNA processing and modification (Appendix A) [22,23].

The expansion of gene families plays a prominent role in increasing phenotypic diversity and facilitating evolutionary adaptation to different environments [24,25]. In this study, we identified a total of 69 and 86 expanded gene families in *C. olor* and *C. atratus*, respectively. To gain insight into their functions, we performed GO enrichment analysis for all expanded genes. Interestingly, we found that the melanogenesis pathway was significantly enriched in *C. atratus*, containing six genes (*MC1R*, *ADCY1*, *ADCY2*, *ADCY3*, *ADCY7*, and *ADCY8*) (Appendix A). The melanogenesis pathway is known to play a critical role in determining the coloration of feathers in birds and other animals, and our findings suggest that this pathway may have undergone positive selection during the evolution of *C. atratus*, potentially contributing to the unique black plumage of this species.

### 2.4. Structural Variants

Structural variations (SVs) wield a potent influence on evolutionary dynamics, complex phenotypic traits, and disease susceptibility [26,27]. In comparison to single nucleotide variants (SNVs) or single nucleotide polymorphisms (SNPs), SVs are known to exert substantially greater genetic effects [27,28]. Thus, in an attempt to elucidate the contribution of SVs to the fascinating plumage coloration patterns that have evolved in *C. olor* and *C. atratus*, we performed a comprehensive SV identification analysis.

In this study, a total of 21,384 structural variations (SVs) were identified in the genomes of *C. olor* and *C. atratus*, ranging in size from 50 to 10,000 bp (Appendix A). Notably, three known genes (*TYR*, *SLC45A2*, and *SLC7A11*) associated with melanin synthesis were found to contain SVs (Figure 2). Specifically, a 608 bp insertion mutation in *C. olor* and a 576 bp insertion mutation in *C. atratus* were identified in the intron region of the *TYR* gene. Additionally, a 519 bp insertion mutation in *C. olor* was discovered in the intron region of the solute carrier family 7 member 11 (*SLC7A11*) gene, which encodes the cystine/glutamate xCT carrier, a key factor for pigment regulation that directly affects the synthesis of pseudomelanin [29]. Moreover, a 1574 bp insertion mutation in *C. atratus* was identified in the intron region of the solute carrier family 45-member 2 (*SLC45A2*) gene.

### 2.5. Positive Selection

The genetic legacy of evolution is imprinted in the genome through selection signals, and their analysis can illuminate the genetic mechanisms driving extreme phenotypic differentiation. Thus, in this study, we conducted selection signal analyses on *C. olor* and *C. atratus* genomes, and a total of 93 biological pathways were enriched (Appendix A), which include anatomical structure development, developmental process, and cellular process, etc. Interestingly, we also discovered several biological pathways, removal of superoxide radicals, and regulation of superoxide metabolic process, which may be related to the longevity of the *Cygnus* genus. Notably, our analysis revealed a positively selected gene (*p* < 0.01), the biogenesis of the lysosomal organelles complex 1 subunit 2 (*BLOC1S2*) gene, which is essential for the normal biogenesis of lysosome-related organelles (LROs), such as melanosomes [30].

### 2.6. Genes with Amino Acid Replacements

Phenotypic variations can be attributed to amino acid mutations to a certain extent. In this study, we conducted a sequence alignment of orthologous genes related to the melanin pathway. We utilized protein sequences from *C. olor*, *C. atratus*, and other homologous species from NCBI and Ensemble databases to perform this analysis. Our findings revealed 15 unique amino acid replacements in five different proteins encoded by the *MC1R*, *TYRP1*, *CORIN*, *SLC45A2*, and *PWWP2A* genes (Figure 3). Specifically, we identified seven amino acid replacements in the *MC1R* gene and two amino acid replacements in the *PWWP2A*, *TYRP1*, *CORIN*, and *SLC45A2* genes.

Of particular interest among the five genes that we identified is *PWWP2A*, which has not been previously implicated in the known melanin pathway. The *PWWP2A* gene has been found to play a key role in neural crest stem cell migration and differentiation during early development [31]. Melanocytes, which are derived from pluripotent neural crest cells [6], undergo several critical developmental stages, including neural crest cell induction, separation, characterization, and migration [32], with the latter two stages being particularly important in the formation of melanocytes. Notably, early investigations of avian embryo neural crest cells have demonstrated that these cells can be identified as melanin precursor cells before they even migrate out of the neural tube melanoblast [33]. Our findings thus suggest the possibility of a previously unrecognized gene related to melanogenesis.

### 2.7. Relationship between PWWP2A Mutation and Melanin Deposition in Zebrafish

Zebrafish, in contrast to model organisms like mice, manifest a heightened proclivity for reproduction. Paramount to their utility is the remarkable transparency of zebrafish embryos, allowing for the development at the granularity of a singular cell. This unique attribute facilitates the real-time imaging of embryonic processes, thereby positioning zebrafish as an exceptional model organism for advancing melanin research. Henceforth, the experimental framework of this investigation was structured around the selection of zebrafish as the primary organism for scrutinizing phenomena such as melanin deposition.

To delineate the spatiotemporal expression pattern of *PWWP2A* during embryogenesis, we conducted a whole-mount in situ hybridization analysis. Our findings reveal that *PWWP2A* exhibited ubiquitous expression during the early developmental stages up to 72 hpf (hours post-fertilization), followed by significantly heightened expression in discrete regions of the brain, including the optic tectum, diencephalon, cerebellum, and midbrain (Figure 4A).

To further elucidate the role of *PWWP2A*, we synthesized sgRNAs in vitro with T7 RNA polymerase (NEB) to generate mutant zebrafish. The genome of the zebrafish was then extracted, amplified, and sequenced to confirm the success of the editing. The sequencing results reveal the successful construction of the mutant zebrafish (*PWWP2A^+/−^*), indicated by the emergence of double peaks from the 17th base of the target (Figure 4B). This mutation caused a frameshift after the target site, leading to the truncation of the presumptive protein. The relative mRNA expression level was significantly reduced in homozygous *PWWP2A*^+/−^ embryos (Figure 4C) (*p* < 0.05). To investigate the function of *PWWP2A* in melanogenesis, we also examined the mRNA expression of classical melanin pathway genes in mutant and wild-type zebrafish. Our results show that, at 24 hpf, the mRNA expression levels of *TYRP1a*, *TYRP1b*, *TYR*, and *DCT* in the mutant zebrafish were significantly higher than those in the wild-type zebrafish (*p* < 0.05). However, at 48 hpf, the mRNA expression levels of *TYRP1a* and *TYR* in mutant zebrafish were notably lower than those in wild-type zebrafish (*p* < 0.05), and there was no significant difference for *TYRP1b* and *DCT*. Similarly, at 72 hpf, the mRNA expression levels of *TYRP1a*, *TYRP1b*, *TYR*, and *DCT* in mutant zebrafish were considerably lower than those in wild-type zebrafish (*p* < 0.05). Overall, our findings suggest that the *PWWP2A* gene may modulate melanin synthesis by downregulating the expression of certain melanin pathway genes.

Subsequently, we performed long-term tracking observations of the *PWWP2A*^+/−^ zebrafish, which revealed that these fish were viable and able to develop into fertile adults, as demonstrated in Figure 4D–F. Notably, our findings differ from those of a previous study, which reported severe head development defects in frog embryos resulting from *PWWP2A* gene knockout [34]. We compared the phenotypes of mutant and wild-type zebrafish at 36 hpf, 48 hpf, and 72 hpf by dividing dozens of zebrafish from the same batch into three groups based on their lateral line melanin content, categorized as low, medium, and high melanin levels. In general, the melanin synthesis in the *PWWP2A*^+/−^ mutant zebrafish was lower than that of the wild-type zebrafish, although the difference between these two groups was not statistically significant.

## 3. Discussion

In this study, firstly, we successfully assembled two high-quality reference genomes for *C. olor* and *C. atratus* using a combination of Nanopore and Illumina sequencing technologies. Our research demonstrates the power of state-of-the-art technologies such as Nanopore sequencing in producing high-quality genomes that can advance our knowledge of biological systems. Moreover, the availability of these reference genomes will serve as a valuable resource for comparative functional and genetic studies of the drastic trait differences observed within the genus *Cygnus*.

The genetic basis of plumage and coat color variation has been a long-standing research interest, with pigmentation being the primary focus of such studies. Vertebrate melanization exhibits remarkable variability in pattern and hue, frequently shaped by intense natural or sexual selection and often understood in terms of genetic regulation [35]. Genetic variation constitutes a crucial source of biological heredity and phenotypic diversity, serving as the raw material for both natural and artificial selection. Genomic variations, including single nucleotide polymorphisms (SNPs), insertions/deletions (INDELs), CNV and structural variations (SVs), provide an indispensable basis for the development of molecular markers and functional gene maps. In this study, we comprehensively screened different types of genetic variations underlying plumage color in order to advance our understanding of this complex trait.

Through positive selection analysis, we enriched a total of 93 biological pathways, including anatomical structure development, developmental process, cellular process, etc. The anatomical structure development pathway may be related to the body shape differentiation and strong flight ability. Some pathways such as the removal of superoxide radicals and the regulation of the superoxide metabolic process may be related to the longevity of the *Cygnus* genus. According to reports, a swan on the island of Zealand in Denmark has lived for at least 40 years. Moreover, we retrieved the *BLOC1S2* gene. The BLOC1S2 protein, a crucial constituent of the BLOC-1 complex, is essential for the normal biogenesis of lysosome-related organelles (LROs), such as melanosomes [30]. Specifically, it has been demonstrated that the pigment-cell-specific cuproenzyme tyrosinase is not efficiently loaded with copper within the trans-Golgi network of mouse melanocytes, and thus, requires reloading with copper within specialized organelles called melanosomes to catalyze melanin synthesis [36]. The supply of copper to melanosomes is facilitated by ATP7A, a protein that localizes to melanosomes in a BLOC-1-dependent manner. As ATP7A targeting melanosomes requires only BLOC-1, the *BLOC1S2* gene may play a crucial role in melanin synthesis.

Through our investigation, we discovered three known genes, namely *TYR*, *SLC45A2*, and *SLC7A11*, which harbor structural variations that have a direct impact on melanin synthesis. The SVs detected in the *TYR* and *SLC7A11* genes were found to have lengths ranging from 500 to 600 bp. This deletion of the *SLC7A11* gene leads to the loss of xCT carrier protein function, causing a reduction in glutathione synthesis, and consequently, blocking the pseudomelanin synthetic pathway. As a result, there is an increase in the production of eumelanin in melanocytes. However, the most significant SV was observed in the *SLC45A2* gene, wherein a large deletion resulted in an SV with a length of 1574 bp, which functions as a transporter on the melanosome membrane to regulate melanin synthesis [37,38].

Moreover, through comparative genomics analysis, we thoroughly screened several genes with amino acid replacements, such as *MC1R*, *TYRP1*, *CORIN*, *SLC45A2*, and *PWWP2A*. Notably, *MC1R* and *TYRP1* have been extensively researched for their significant impact on melanin synthesis [11,12,13,39,40,41,42]. Additionally, the *Corin* gene, previously identified in the human heart, encodes a unique mosaic serine protease [43]. Recent findings indicate its expression in the dermal papilla of all pelage hair follicle types, where it modulates the balance between eumelanogenesis and pheomelanogenesis [44]. By regulating ASIP activity, it actively participates in the regulatory pathway of melanin synthesis and consequently influences the formation of tiger hair color [45]. Similarly, the *SLC45A2* gene encoding a transporter mediating melanin synthesis-related protein plays a crucial role in melanophore development in the Japanese medaka [46,47,48]. The latest research on the Japanese population also suggests that a 4-bp deletion variant in the promoter region of the *SLC45A2* gene contributes to hair color variation [49]. Remarkably, our study unraveled a novel gene (*PWWP2A*) associated with melanogenesis. Notably, previous studies on melanin pathway genes have not considered the role of *PWWP2A*. It may assume a crucial function in the formation of melanocytes by influencing the migration and differentiation of neural crest stem cells.

To investigate the potential role of the *PWWP2A* gene in melanin synthesis, we performed whole-mount in situ hybridization, RT-PCR, and CRISPR/Cas9 experiments. Although the results suggest that *PWWP2A* may be involved in melanin synthesis, the difference between the treatment group and control group was not statistically significant, possibly because the heterozygous *PWWP2A*^+/−^ allele has a minor effect on the phenotype. Similar results were observed for the *BLOC1S1* mutant zebrafish, wherein heterozygous mutants did not show a discernible phenotype, while homozygous mutants displayed an observable phenotype at 3 days post-fertilization (dpf) [50]. Additionally, the regulation of plumage coloration is a complex process controlled by multiple genes of minor effect, with up to 688 genes involved [51] (last update: 8 December 2022). Therefore, the genetic effect of *PWWP2A* on plumage coloration may not be as prominent as initially thought. Furthermore, the biological function of proteins is closely related to their three-dimensional structure [52], and it is unclear whether the deletion of the *PWWP2A* gene results in changes to the three-dimensional structure of its protein. To further investigate the genetic effect of the *PWWP2A* gene on melanin synthesis, future studies should aim to obtain homozygous mutants by crossing *PWWP2A*^+/−^ heterozygous individuals and analyze their phenotypes. Additionally, exploring the three-dimensional structure of the mutant protein could provide valuable insights into its biological functions.

Melanin, a dark brown pigment ubiquitous in various bacterial, fungal, plant, and animal species, is synthesized within specialized organelles called melanosomes [53]. The biosynthesis of melanin is meticulously governed by a multifaceted interplay of intracellular signaling cascades, among which the most commonly implicated ones are the MC1R/α-MSH [54], cAMP/PKA [55], MAPK/ERK [56], PI3K/Akt [57], and Wnt/β-catenin [58] signaling pathways. Nonetheless, the synthesis of melanin is a complex and polygenic trait, regulated by a plethora of genes with modest effects, as evidenced by studies reporting the involvement of as many as 688 genes [51] (last update: 8 December 2022). In the present study, a comprehensive investigation of genetic variations in plumage color was performed using various approaches, including the screening of six expanded genes (*MC1R*, *ADCY1*, *ADCY2*, *ADCY3*, *ADCY7*, and *ADCY8*), three genes with structural variations (*TYR*, *SLC45A2*, and *SLC7A11*), one positively selected gene (*BLOC1S*2), and five genes with amino acid replacements (*MC1R*, *TYRP1*, *CORIN*, *SLC45A2*, and *PWWP2A*). Of particular note, our study identified *PWWP2A* as a novel gene that is linked to melanogenesis. Drawing upon the collective findings of this study and previous research, a signal pathway diagram was generated to represent melanin synthesis (Figure 5). In comparison to previously published signal pathway diagrams [59,60], we have annotated the key genes identified in this study at their corresponding node positions in the melanin synthesis pathway. Furthermore, we have extended the melanin pathway to the upstream neural crest cells for the first time, thereby enhancing our understanding of the melanin synthesis signaling pathway.

## 4. Materials and Methods

### 4.1. Sample Collection and Genomic Sequencing

Blood specimens were procured from one adult female *C. olor* (approximately 5.5 years old) and one adult female *C. atratus* (approximately 6 years old) individuals, housed at the Kunming Wildlife Zoo in Yunnan Province, China. Genomic DNA of high integrity was isolated from the whole blood utilizing the conventional cetyltrimethylammonium bromide (CTAB) extraction method. An Agilent 2100 bioanalyzer (Agilent Technology Co., Ltd., Santa Clara, CA, USA) was employed to assess the quality of the isolated DNA. All the experimental procedures were approved by the Animal Care and Use Committee of Yunnan Agricultural University (Approval Code: 202210003, Approval Data: 30 October 2022).

For short-read sequencing by Illumina platform, 100 ng of genomic DNA (gDNA) was employed for library preparation. Genomic DNA was sonicated to 350 bp fragments using an ultrasonicator, and the library was generated with the NEBNext Ultra DNA library prep kit (Shanghai Jinpan Biotechnology Co., Ltd, Shanghai, China) following the manufacturer’s guidelines. Libraries were sequenced on an Illumina HiSeq X Ten sequencer in 2 × 150 bp paired-end mode.

For long-read sequencing by Oxford Nanopore, gDNA was prepared utilizing the NEB Next FFPE DNA Repair Mix kit (Shanghai Jinpan Biotechnology Co., Ltd, Shanghai, China) and subsequently processed with the ONT Template prep kit (SQK-LSK109, UK) as per the manufacturer’s instructions. The final product was quantified using a Qubit fluorometer (Thermo Fisher Scientific, Waltham, MA, USA) and loaded onto an R9 flow cell for sequencing on the ONT PromethION platform, employing the corresponding R9 cell and ONT sequencing reagent kit (Oxford Nanopore Technologies Ltd., Oxford, UK) in accordance with the manufacturer’s guidelines.

### 4.2. Genome Size Estimation

Illumina short reads were processed using fastp v0.23.4 [61] with parameters “−q 10 −u 50 −y −g −Y 10 −e 20 −l 100 −b 150 −B 150.” The genome size was approximated employing the K-mer method on clean reads via Jellyfish v2.3.0 software [62]. The K-mer number was set to 21, and the genome size was determined using the following formula: Genome size = k*_num_*/k*_depth_*, where k*_num_* represents the K-mer number and k*_depth_* is the overall depth estimated from the K-mer distribution.

### 4.3. Genomic Assembly and Evaluation

Utilizing Canu v1.4 software, Nanopore third-generation sequencing data underwent correction to achieve high-accuracy results [63]. Subsequently, Wtdbg2 v2.2 software facilitated genome assembly based on the corrected sequencing data [64]. The third- and second-generation sequencing data were subjected to three rounds of correction employing Racon and Pilon v1.23 software, respectively [65,66]. The assembled genomes were assessed through the following methods: (1) Coverage rate of second-generation sequencing reads. The integrity of the assembled genomes was appraised by mapping short sequences derived from the second-generation high-throughput sequencing to the assembled genomes using BWA v0.7.17 software [67]. (2) Core gene integrity (employing CEGMA (v2.5)). The final genome assemblies were evaluated for integrity, consisting of 458 core genes conserved among eukaryotes [68]. (3) BUSCO assessment (with BUSCO (v2)), the completeness of the genome assemblies was assessed based on the OrthoDB (http://cegg.unige.ch/orthodb) database (accessed on 23 October 2022), encompassing 2586 conserved core genes [69].

### 4.4. Genome Annotation

The prediction of repeat sequences employed de novo-based and homology-based approaches, utilizing LTR_FINDER and RepeatScout v1.0.6 software [70,71] with default parameters to establish a de novo repeat sequence library. The database was classified by PASTEClassifier [72], merged with Repbase [73], and repeat sequences were predicted using the newly constructed repeat library database through RepeatMasker v3.1.9 software [74].

Protein-coding gene annotation in the genomes integrated de novo prediction and homologue-based prediction. Ab initio prediction was executed using GENSCAN v1.0 [75], Augustus v2.4 [76], GlimmerHMM v3.0.4 [77], GeneID v1.4 [78], and SNAP v2006-07-28 [79] with default parameters. Homology-based prediction, referencing four species (*Anas platyrhynchos*, *Meleagris gallopavo*, *Gallus gallus*, and *Anser cygnoides*), was conducted using GeMoMa v1.3.1 [80,81]. EVM (v1.1.1) integrated the prediction outcomes [82].

For noncoding RNA prediction, Infernal (v1.1) [83] identified microRNA and rRNA based on the Rfam database [84]. while tRNAscan-SE [85] recognized tRNA. Protein-coding gene annotation was performed using BLAST v2.2.31 (1 × 10^−5^) [86] against the NR [87], KOG [88], GO [89], KEGG [90], and TrEMBL [91] functional databases.

### 4.5. Comparative Phylogenomics

Gene family clustering utilized protein sequences from *Anas platyrhynchos*, *Cairina moschata*, *Anser cygnoides*, *Anser brachyrhynchus*, *Gallus gallus* (from Ensembl Release 102), *Chauna torquata*, *Anseranas semipalmata*, *Oxyura jamaicensis*, *Aythya fuligula* (from NCBI), *C. olor*, and *C. atratus*. The transcript was translated into an amino acid sequence, followed by an all-vs.-all comparison using BLAST-2.2.26 with parameters “-p blastp -m 8 × 10^−10^ -F F”. OrthoMCL v2.0 software [92] classified gene families, and one-to-one single-copy orthologous genes were aligned by Muscle (v3.8.31) [93]. An eleven-species phylogenetic tree was constructed using the maximum-likelihood method in PhyML (v3.0) software with “-d nt -b 100 -m HKY85 -a e -t e” [94] and RAxML-8.2.12 software, concatenating all four-fold degenerate sites and coding sequences of single-copy orthologues. Divergence times were estimated via the MCMCTree program in the PAML package (v4.9) [95] with “clock = correlated rates, model = JC69, burnin = 20,000, and nsample = 100,000”, and calibration time was obtained from the TimeTree database (http://www.timetree.org/) (accessed on 26 January 2023) [96]. Expansion and contraction were detected using CAFE (v4.2.1) [97]. The *C. olor* and *C. atratus* genomes were aligned with MUMmer3.23 software (https://mummer.sourceforge.net/) (accessed on 30 January 2023), and then structure variations (SVs) were identified by Assemblytics [98]. Miropeats (v2.02) [99] and Adobe Illustrator v24.0 software were used to visualize the SVs.

### 4.6. Whole-Mount In Situ Hybridization (WISH)

Wild-type zebrafish (*Danio rerio*) (AB strain) were procured from a zebrafish supplier and maintained under standard conditions. Embryos, obtained through natural spawning, were preserved at 28 °C. PCR amplification of templates involved adding a T7 promoter sequence to the primers (primers listed in Appendix A). RNA probes, generated by in vitro transcription, were labeled with DIG RNA Labeling Mix (Roche). Embryos were fixed overnight at 4 °C in 4% paraformaldehyde, and the egg membrane was carefully removed using forceps. Exposed embryos were dehydrated in a graded methanol series (25%, 50%, 75%, 100%) and stored at −20 °C. Prior to WISH, embryos were rehydrated in a series of methanol solutions diluted with PBST, allowing for 5 min per step. Embryos underwent proteinase K (10 µg/mL) digestion for 5 min, were washed with PBST twice, refixed with 4% PFA in PBST for 20 min, washed with PBST (5 × 5 min), and equilibrated in hybridization solution (HYB−) overnight at 67.5 °C. Subsequently, embryos were incubated in 200 µL hybridization solution (HYB+) containing the probe overnight at 67.5 °C. Washing steps at 67.5 °C involved 50% formamide diluted with 2 × SSCT (2 × 30 min), 2 × SSCT (15 min), 0.2 × SSCT (2 × 30 min), and MABT (2 × 5 min at room temperature). Embryos were then blocked in blocking solution for 4 h at room temperature, incubated in 1:5000 anti-DIG-AP Fab fragments (Roche) overnight at 4 °C, washed with 10% sheep serum in MABT for 25 min, and repeatedly washed with MABT (8 × 1 h). Following this, embryos were washed three times with 1 × PBS and twice with ddH_2_O. Incubation with BM Purple AP Substrate (Roche) occurred in the dark. Upon signal observation, the reaction was halted using 1 × PBS, and a series of glycerine solutions (30%–50%–80%) was employed. Embryos were photographed and stored in 80% glycerine for extended periods at 4 °C.

### 4.7. Generation of Mutant Fish

*PWWP2A* mutant lines were produced using the CRISPR/Cas9 system, following established methodologies [100]. Target sites were selected via the SYNTHEGO and CHOPCHOP design websites (https://www.synthego.com/, http://chopchop.cbu.uib.no/, with primers listed in Appendix A) (accessed on 17 April 2023). Templates for guide RNAs were PCR-amplified, incorporating a T7 promoter and gRNA scaffold sequence, and sgRNAs were synthesized in vitro using T7 RNA polymerase (NEB). Subsequently, sgRNAs were precipitated with isopropanol/sodium acetate. A mixture of 1.4 nL EnGen^®^ Spy Cas9 NLS (NEB) and sgRNA was injected into fertilized wild-type eggs in appropriate proportions.

### 4.8. RT-PCR

Total RNA was isolated from pooled zebrafish embryos at 24, 48, and 72 hpf (15 samples per group), utilizing TRIzol reagent (Thermo Fisher Scientific, State of Delaware, USA). Zebrafish embryo cDNA synthesis was achieved with the PrimeScript™ RT Reagent Kit with gDNA Eraser (TaKaRa Bio Inc., Tokyo, Japan). Subsequently, TB Green^®^ Premix Ex Taq™ (Takara) was employed for RT-PCR. The used primers can be found in Appendix A.

## 5. Conclusions

In conclusion, the present study provides valuable insights into the genomic features of two swan species, *C. olor* and *C. atratus*. Our results suggest that these two species diverged from their common ancestor approximately 7.7–13.8 Mya. By performing genome-wide comparative analyses, we identified several genes that may be related to plumage color, including one positively selected gene, five genes with amino acid replacements, and three genes with structural variants. Of particular interest, a novel gene (*PWWP2A*) was discovered, and our targeted gene modification analysis in zebrafish using the CRISPR/Cas9 system provides evidence for its potential role in melanin synthesis. These findings expand our knowledge of the genetic basis underlying plumage color in swans. Furthermore, the high-quality genome references generated in this study will serve as valuable resources for future comparative functional and genetic studies aimed at unraveling the mechanisms underlying drastic trait differences and evolution within the genus *Cygnus*.

## Figures and Tables

**Figure 1 ijms-24-16953-f001:**
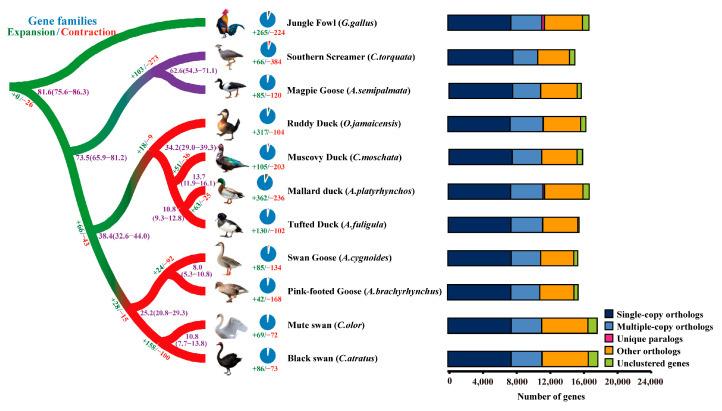
Phylogenetic analysis of *C. olor* and *C. atratus* with other species. *G. gallus* was used to root the tree. The purple numbers are the predicted divergence times (Mya). Branches including the *Galliformes*, *Anatidae* of *Anseriformes*, and *Anseranatidae* and *Anhimidae* of the *Anseriformes* clade are highlighted in green, red, and purple, respectively. The number on each branch represents the expanded (green, positive) and contracted (red, negative) orthologue groups. The right side of this figure represents the OrthoMCL clusters of *C. olor* and *C. atratus* and other species.

**Figure 2 ijms-24-16953-f002:**
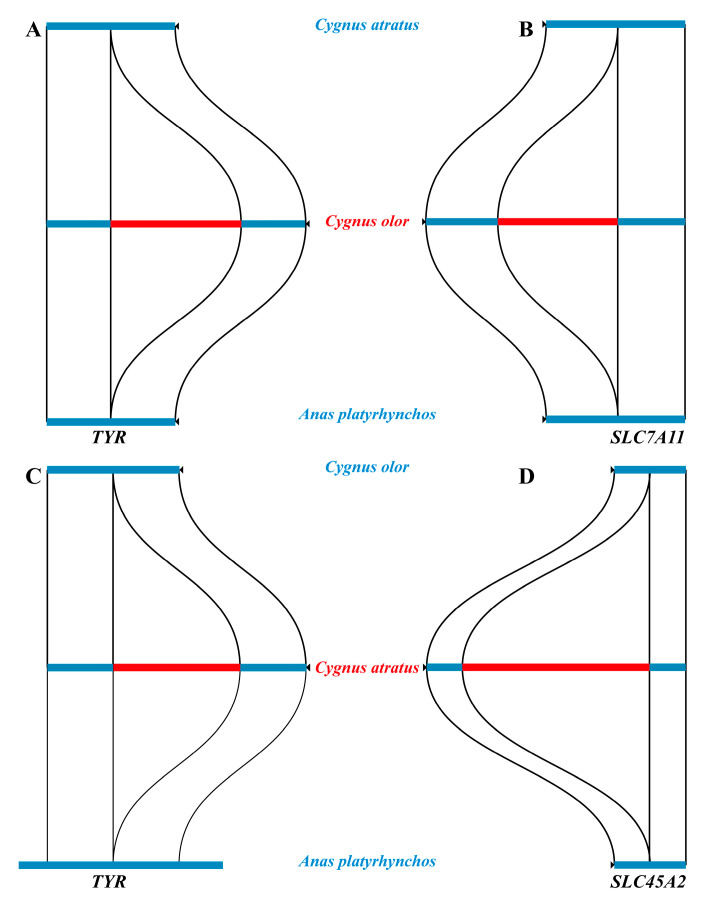
Schematic diagram of SVs of the *TYR*, *SLC45A2*, and *SLC7A11* genes in *Cygnus olor*, *Cygnus atratus*, and *Anas platyrhynchos*. (**A**) the sequence alignment for *TYR* gene between *Cygnus atratus* and *Anas platyrhynchos*. (**B**) the sequence alignment for *SLC7A11* gene between *Cygnus atratus* and *Anas platyrhynchos*. (**C**) the sequence alignment for *TYR* gene between *Cygnus olor* and *Anas platyrhynchos*. (**D**) the sequence alignment for *SLC45A2* gene between *Cygnus olor* and *Anas platyrhynchos*.

**Figure 3 ijms-24-16953-f003:**
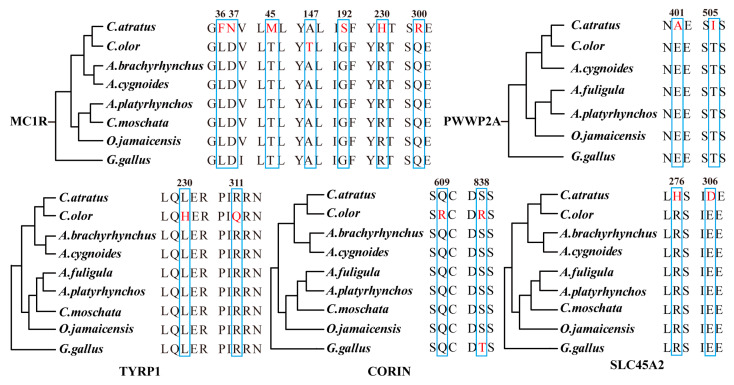
The amino acid replacements of key genes in *Cygnus* genomes. The amino acid replacements are shown in red and marked with boxes.

**Figure 4 ijms-24-16953-f004:**
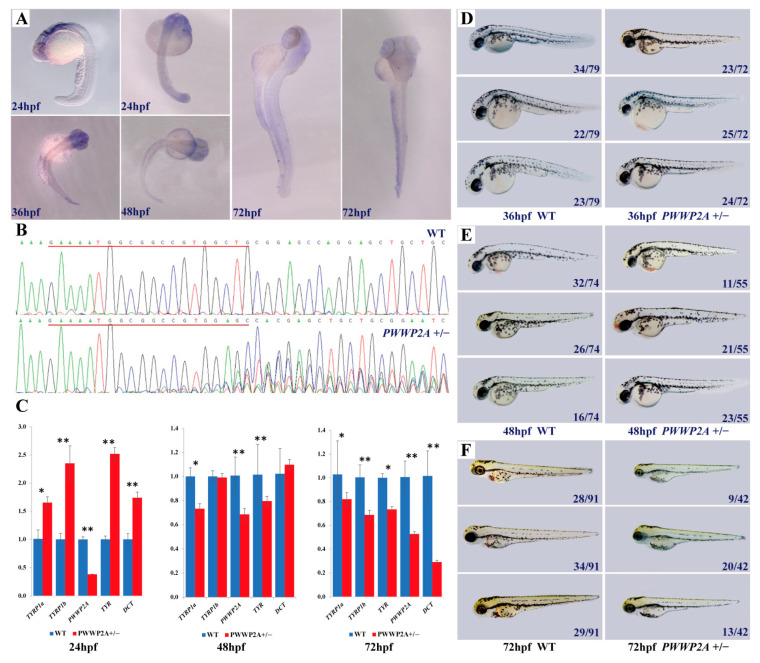
The relationship between *PWWP2A* mutation and melanin deposition in zebrafish. (**A**) Detection of the *PWWP2A* transcript by whole-mount in situ hybridization. hpf, hours post-fertilization. (**B**) Sequencing results of the *PWWP2A* gene of wild-type and mutant zebrafish. (**C**) Relative mRNA levels of *PWWP2A* in WT and heterozygous mutants were assayed by RT-PCR at 24, 48, and 72 hpf. The results are expressed as the mean ± SEM of three independent experiments (* *p* < 0.05, ** *p* < 0.01; *t* test). (**D**–**F**) Phenotypic comparison of mutant zebrafish and wild-type zebrafish at 36, 48, and 72 hpf. Dozens of zebrafish in the same batch were divided into three groups according to the melanin content of the lateral line, and the number in the figure represents the number of zebrafish in each group.

**Figure 5 ijms-24-16953-f005:**
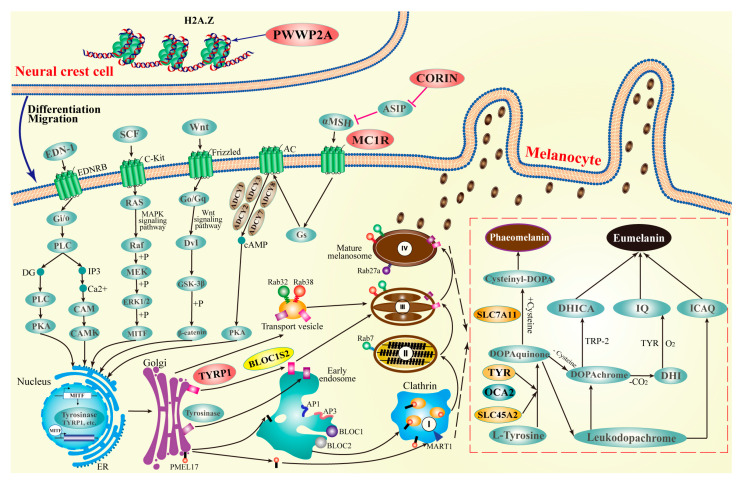
The signal path diagram for melanin synthesis. The biological process from the differentiation and migration of neural crest cells to melanin synthesis is represented. The brown, orange, yellow, and red circles represent the expanded genes, genes with structural variants, positively selected genes, and genes with amino acid substitutions, respectively.

**Table 1 ijms-24-16953-t001:** Genomes assembly statistics.

Species	Contig Number	Contig Length (Mb)	Contig N50 (Mb)	Contig N90 (Mb)	Contig Max (Mb)	GC Content (%)	Complete BUSCOs
*C. olor*	335	1124.00	26.82	4.90	77.28	42.04	2546 (98.45%)
*C. atratus*	469	1129.04	21.91	4.55	65.49	42.06	2542 (98.30%)

## Data Availability

All relevant data are within the paper and its Appendix A files. The genome assembly and raw sequencing data have been deposited in the China National GeneBank DataBase with the project ID PRJCA013296, which includes the genome assembly of *C. olor* (SAMC1014187) and *C. atratus* (SAMC1014188), Nanopore sequencing data of *C. olor* (SAMC1016507) and *C. atratus* (SAMC1016508), and Illumina sequencing data of *C. olor* (SAMC1016505) and *C. atratus* (SAMC1016506).

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
