# Peer review of "Two High-Quality Cygnus Genome Assemblies Reveal Genomic Variations Associated with Plumage Color"

_ijms, 2023, doi:10.3390/ijms242316953_

Round 1

Reviewer 1 Report

Comments and Suggestions for Authors

In my opinion the study provides an interesting novel evidence for the genetic background of melanine pigmentation in swans, which might be universal patterns in vertebrates, as confirmed by the experiments with zebra finches. The study is clear and convincing, and presents novel possibilities to expand our knowledge of the genetic determination of such traits in animals as colouration and longevity. Thus, the study is valuable contribution to the topic. My minor concerns are that the melanin pigmentation is not the only source of colouration of feathers and hair, so I suggest more cautious phrasing on the 'predominant' role of melanin in colouration. I suggest more thorough presentation of scientific names of the presented species, and more clear labelling of FIgure 1, to improve readability for a reader not familiar with bird taxonomy. But these are just minor editorial comments, which I marked in the attached msc. I complement the authors for the interesting and valuable paper.

Reviewer 2 Report

Comments and Suggestions for Authors

Implementation

Cygnus atratus is a distinctive species with plumage contrasting with its closely related northern white swan. The Locus MC1R (melanocortin-1 receptor) underlies the intraspecific variability of melanin-based dark plumage in several unrelated birds with plumage polymorphism. There is a highly significant relationship between MC1R variability and plumage in swans (Cygnus), which show extreme differences in melanic plumage phenotypes across species (from white to black). Reconstruction of MC1R evolution on newly generated independent molecular phylogenesis of Cygnus and related genera shows that these supposed melanizing mutations independently derived from two melanic lines.

The aim of this study was to gain a deeper understanding of the genetic mechanisms driving phenotypic differences in plumage colour in Cygnus species.

The title of the work reflects the content of the work

Abstract

The abstract contains the most important results obtained in the study, I have no comments on this part.

Introduction:

The introduction to the subject of research and the selection of literature is appropriate.

It is worth putting a research hypothesis at the end and re-editing requires the objective of the work

Material and Methods:

The research methods used are modern and suitable for this type of research. Please specify and supplement the methodology with information on how many birds of each species were sampled and their age.

Results:

The results are presented in a comprehensible way and are presented in the form of 1 table and 4 graphs with their description. This chapter is well written and unobjectionable.

Discussion:

The discussion was conducted fairly. The authors referred to all the results obtained in the study. Relevant literature has been cited.

Final remark:

The submitted work for evaluation is of a very high scientific standard and very well edited. It should be noted that the authors have a great deal of knowledge in the field of genomic research. The results provide new information on the genomic characteristics of two swan species, C. olor and C. atratus. Of particular interest is the discovery of a new gene (PWWP2A). These findings expand our understanding of the genetic basis of plumage colour in swans. The work is most suitable for printing in IJMS.

Reviewer 3 Report

Comments and Suggestions for Authors

The present study provides valuable insights into the genome features of two swan species - Cygnus olor and Cygnus atratus. The results of this study allowed us to explain the molecular basis of the different plumage colors of Cygnus olor (white plumage) and Cygnus atratus (black plumage). The article presents a two-quality Cygnus genome procured using Illumina and Nanopore technologies. A novel gene, PWWP domain containing 2A (PWWP2A), has been identified that is related to plumage color for the first time.

General concept comments:

In my opinion, the article should be supplemented with the following information:

In the Introdaction chapter

The role of diverse plumage colors in nature.

Different colors of males and females in polygamous birds and similar in monogamous ones

Feather function

A brief description of both species, place of occurrence, body weight of adult swans, and differences in BW

What dyes influence the individual colors of plumage (feathers)

In the Results section

I suggest moving Figure S3 and S4 to the Results chapter

In the Material and methods section

How many birds were assessed and what was their gender?

Editorial notes

Please prepare the article in accordance with the instructions for authors:

• For significance please use lowercase "p" in italic instead of uppercase "P" throughout the main article

• In References chapter please use a "dot" after each abbreviation, for example Sci. Rep. instead of Sci Rep

Abbreviated name journal, item 11, 15, 16, 20, 23, 28, etc

for page ranges use long (-) from the symbol function, instead of short (-) from the keyboard
